# What is the recovery rate and risk of long-term consequences following a diagnosis of COVID-19? A harmonised, global longitudinal observational study protocol

Louise Sigfrid  ,[1] Muge Cevik,[2] Edwin Jesudason,[3] Wei Shen Lim,[4] Jordi Rello  ,[5,6] John Amuasi,[7] Fernando Bozza,[8] Carlo Palmieri,[9,10] Daniel Munblit  ,[11,12] Jan Cato Holter,[13,14] Anders Benjamin Kildal  ,[15] Luis Felipe Reyes,[16] Clark D Russell,[17] Antonia Ho,[18] Lance Turtle,[19,20] Thomas M Drake,[21] Anna Beltrame,[22] Katrina Hann,[23] Ibrahim Richard Bangura,[24] Robert Fowler,[25] Sulaiman Lakoh,[23] Colin Berry,[26] David J Lowe,[27] Joanne McPeake  ,[28,29] Madiha Hashmi,[30] Anne Margarita Dyrhol-Riise,[31] Chloe Donohue,[32,33] Daniel Plotkin,[34] Hayley Hardwick,[32,33] Natalie Elkheir,[35] Nazir I Lone,[36] Annemarie Docherty,[37] Ewen Harrison,[37] J Kenneth Baille,[38] Gail Carson,[1] Malcolm G Semple,[39,40] Janet T Scott[41]

For numbered affiliations see end of article.

**Correspondence to**
Dr Louise Sigfrid;
louise.sigfrid@gmail.com

## ABSTRACT

**Introduction** Very little is known about possible clinical sequelae that may persist after resolution of acute COVID-19. A recent longitudinal cohort from Italy including 143 patients followed up after hospitalisation with COVID-19 reported that 87% had at least one ongoing symptom at 60-day follow-up. Early indications suggest that patients with COVID-19 may need even more psychological support than typical intensive care unit patients. The assessment of risk factors for longer term consequences requires a longitudinal study linked to data on pre-existing conditions and care received during the acute phase of illness. The primary aim of this study is to characterise physical and psychosocial sequelae in patients post-COVID-19 hospital discharge.

**Methods and analysis** This is an international open-access prospective, observational multisite study. This protocol is linked with the International Severe Acute Respiratory and emerging Infection Consortium (ISARIC) and the WHO's Clinical Characterisation Protocol, which includes patients with suspected or confirmed COVID-19 during hospitalisation. This protocol will follow-up a subset of patients with confirmed COVID-19 using standardised surveys to measure longer term physical and psychosocial sequelae. The data will be linked with the acute phase data. Statistical analyses will be undertaken to characterise groups most likely to be affected by sequelae of COVID-19. The open-access follow-up survey can be used as a data collection tool by other follow-up studies, to facilitate data harmonisation and to identify subsets of patients for further in-depth follow-up. The outcomes of this study will inform strategies to prevent long-term consequences; inform clinical management, interventional studies, rehabilitation and public health management to reduce overall morbidity; and improve long-term outcomes of COVID-19.

**Ethics and dissemination** The protocol and survey are open access to enable low-resourced sites to join the study to facilitate global standardised, longitudinal data collection. Ethical approval has been given by sites in Colombia, Ghana, Italy, Norway, Russia, the UK and South Africa. New sites are welcome to join this collaborative study at any time. Sites interested in adopting the protocol

## Strengths and limitations of this study

► This is an open-access protocol and data collection forms to facilitate standardised, multisite data collection to forward knowledge into long-term consequences of COVID-19.

► This study aims to inform strategies to prevent longer term sequalae; inform clinical management, rehabilitation and public health management strategies to reduce morbidity; and improve outcomes.

► The protocol will be used to follow-up a subset of patients, already included in the existing International Severe Acute Respiratory and Emerging Infection Consortium (ISARIC) cohort of more than 95 966 individuals hospitalised with confirmed COVID-19 across 42 countries (as of 20 November 2020).

► The follow-up data will be linked with acute phase data already documented using the ISARIC/WHO standardised Core or RAPID case report forms.

► The data collection tool is developed to facilitate wide dissemination and uptake with limited resources to mitigate resource limitations during the pandemic.

as it is or in an adapted version are responsible for ensuring that local sponsorship and ethical approvals in place as appropriate. The tools are available on the ISARIC website (www.isaric.org).

**Protocol registration number** osf.io/c5rw3/
**Protocol version** 3 August 2020
**EuroQol ID** 37035.

## INTRODUCTION

COVID-19, caused by SARS-CoV-2 infection, can lead to a diverse range of clinical manifestations, ranging from an asymptomatic infection to an acute respiratory distress syndrome and multiorgan failure with high risk of mortality.[1] It is established that SARS-CoV-2 infects the respiratory tract but that ensuing viral replication and immune response may also affect other organs, which can lead to a risk of heart, renal and liver injury, in addition to an acute systemic inflammatory response and accompanying circulatory shock.[2–4] While most people have uncomplicated recoveries, some have prolonged illness even after recovery from the acute illness.[5–7] Identifying longer term potential consequences and relationship with the acute illness is important for the management of patients, in particular, understanding how these interact and affect those already living with other conditions such as cardiovascular disease and cancer will be paramount.

However, very little is known about possible clinical sequelae that may persist after the resolution of acute infection. A recent longitudinal cohort of 143 patients followed after hospitalisation from COVID-19 in Italy reported that 87% had at least one ongoing symptom, most (55%) with three or more symptoms at 60-day follow-up, fatigue (53%), dyspnoea (43%), joint pain (27%) and chest pain (22%) being the most common. COVID-19 was associated with worsened quality of life among 44% of patients.[6] Prolonged course of illness has also been reported among people with mild COVID-19 who did not require hospitalisation.[5 7 8]

Increasing evidence also suggests that infection with SARS-CoV-2 can cause neurological consequences,[2] including altered mental status, comprising encephalopathy or encephalitis and primary psychiatric diagnoses.[9] While these symptoms arise acutely during the course of infection, less is known about the possible long-term consequences. Severely affected COVID-19 cases experience high levels of proinflammatory cytokines and acute respiratory dysfunction that often require assisted ventilation. These are known factors suggested to cause cognitive decline.[2 10]

Post-traumatic stress disorder (PTSD) and postintensive care syndrome after intensive care unit (ICU) stay has been well documented previously.[11–13] A systematic review of consequences after hospitalisation or ICU stay for SARS-CoV and Middle East respiratory syndrome coronavirus found sequelae up to 6 months after discharge. Common consequences, besides impaired diffusing capacity for carbon monoxide and reduced exercise capacity were PTSD (39%), depression (33%) and anxiety (30%).[14] Additionally, serial CT scans postdischarge

after SARS-CoV showed a gradual healing of pulmonary injury, with pulmonary consequences lasting more than 6 months postdischarge.[15]

Early indications suggest patients with COVID-19 will need even more psychological support than typical post-ICU patients because of higher levels of 'survivors' guilt' and PTSD.[16] In addition, the characteristics of the initial cellular immune and antibody response to SARS-CoV-2 have not been fully defined, and it is not known if the immune responses generated by infection provides long-term protective immunity. Identifying multidisciplinary sequalae and complications through high-quality, global studies throughout the course of COVID-19 is important for the acute and longer term management of patients.[4 17] The emerging data and anecdotal evidence of long-term recovery and persistent debilitating symptoms highlight the need for robust, standardised studies to assess the risk of and risk factors for COVID-19 sequalae.

The purpose of this study is to establish a longitudinal cohort of patients with COVID-19 to characterise the risk of long-term consequences over time in different populations globally.[18] The primary outcome is to characterise physical and psychosocial consequences in patients post-COVID-19 infection. Secondary outcomes include estimating the risk of and risk factors for post-COVID-19 medical sequalae, psychosocial consequences and long-term outcomes. The results will inform strategies to prevent long-term consequences; inform clinical management, interventional research and direct rehabilitation; and inform public health management to reduce overall morbidity and improve outcomes of COVID-19. This study is open access for any sites globally to join the collaboration to characterise COVID-19 in different populations. Providing research tools that can be implemented for free in low-resourced settings can increase equity in implementation of and inclusion in clinical research studies.

## METHODS AND ANALYSIS

This protocol and data collection surveys have been developed by the ISARIC COVID-19 global follow-up working group and informed by a wide range of global stakeholders with expertise in clinical research, outbreak research, infectious disease, epidemiology, respiratory, critical care, rehabilitation, neurology, psychology, rheumatology, cardiology, oncology and public health medicine.[18]

### Study design

This is an international prospective, observational multisite cohort study to assess risk of and risk factors for longer term physical and psychosocial consequences of COVID-19. The study conforms to research ethics standards and has been approved by research ethics boards in Colombia, Ghana, Italy, Norway and the UK and submitted for ethics approval in additional sites and

countries. All sites adopting the protocol are responsible for ensuring that they have local ethical approval in place.

## Population and setting

This protocol builds on the ISARIC/WHO COVID-19 Clinical Characterisation Protocol (CCP) and associated data collection forms already in operation, the Core and Rapid case report forms (CRFs).[19] These CRFs were developed to standardised clinical data collection on patients admitted with suspected or confirmed COVID-19, with clinical data on more than 95 966 individuals hospitalised with confirmed COVID-19 infection across 42 countries documented in a central database (as of 20 November 2020).[18 20] These CRFs collect data on demographics, preexisting comorbidities and risk factors, signs and symptoms experienced during the acute phase, and care and treatments received during hospitalisation.[19]

This acute phase data will be linked with the follow-up data to inform analysis. A subset of these patients with confirmed COVID-19 will be followed up over time, documenting data on longer term consequences using the tier 1 follow-up data collection surveys.[18] There is no limit on the number of sites or countries taking part.

New clinical sites are invited to take part in this global collaborative effort and use these open-access tools. New sites can complete the Core or RAPID acute phase CRFs prospectively or retrospectively (figure 1). Sites adopting the protocol and tools for collaborative or independent studies are responsible for ensuring local regulatory and ethical approvals are in place as appropriate.

Specific inclusion and exclusion criteria for the follow-up cohort are as follows:

### Inclusion criteria
► People aged 16 years and older.
► Laboratory or physician confirmed COVID-19.
► At least 1 month postdischarge from hospital or health centre.
► Person (or family member/carer for patients who lack capacity) consent to participate.

### Inclusion of vulnerable participants
The data collection surveys and validated tools are developed for anyone who fit the inclusion criteria, including pregnant women, elderly and those who are immunosuppressed. An aligned study protocol and survey will be developed for following up children.

### Outcomes and procedure
The follow-up protocol and survey are designed to be flexible in a tiered approach to be adapted depending on local resources and research needs. The tier 1 survey is designed to enable patient self-assessment to facilitate distribution to all patients that fit the inclusion criteria, using a range of methods via post, an online link for self-completion or via telephone or in-clinic completion. A combination of methods can be used to be as inclusive as possible and depending on site resources. The tier 1 survey designed can be used to identify people with set

symptoms for further in-depth in clinic analysis (tier 2). The tier 1 survey is open access for sites to adopt as it is or adapt and combine with sampling and diagnostic methods to support further analysis (figure 1).

## Serial follow-up

The aim is to follow-up patients serially over time at regular intervals, for as long as there is a need and resources. The tier 1 initial survey will be used at the inital follow-up time-point (at 1–3 months postdischarge). The tier 1 ongoing survey will be used at subsequent follow-up time points every 3–6 months depending on resources (figure 2). To faciliate combined analysis, the time frames in figure 2 are recommended; however, these can be adapted depending on local resources and needs. For sites that have capacity, the tier 1 form can be used to identify people for more in-depth in-clinic follow-up. By being designed to enable patient self-completion, it allows wide distribution at low resource need. It can also be completed via in-clinic or telephone assessments during check-ups or for patients that are still hospitalised. The module will collect data on demographics, hospital stay and readmissions, all-cause and cause-specific mortality (after the initial index event), specific consequences including: deep vein thrombosis (DVT), pulmonary embolism, recent febrile illness, new or persistent symptoms, quality of life (measured by EQ-5D-5L), dyspnoea (assessed using MRC dyspnoea scale), difficulties in functioning (UN/Washington disability score), lifestyle and socioeconomic data.[18] Medical records will be used to assess if a person is deceased before each follow-up time point and documented in the database. The ISARIC collaborative follow-up study is registered with EuroQol. Sites that wish to adopt the survey for independent studies, outside of the ISARIC collaboration, need to register to use the EQ-5D-5L tool with EuroQol.

## Substudies

The tier 1 follow-up module can be used to identify subsets of patients experiencing specific symptomatology or syndromes for further follow-up (tier 2). By using a tiered approach, additional specialist modules can be added for more complex follow-up of emerging consequences in a flexible, adaptable way, of a subset of patients and combined with sampling and diagnostics (figures 1 and 2).

Serial follow-up will continue beyond 12 months at 3–6 months' interval for up to 3 years depending on resources.

## Biological samples

The tier 1 surveys can be used on its own for data collection or in combination with sampling (eg, respiratory samples, serum, blood, stool and urine), for immunology, pathophysiology, genomics or other studies.[16] This protocol builds on the ISARIC/WHO CCP, which includes an adaptable research sample schedule for sites with resources for research sampling and analysis.[19 21] The CCP is designed to enable a harmonised, multisite

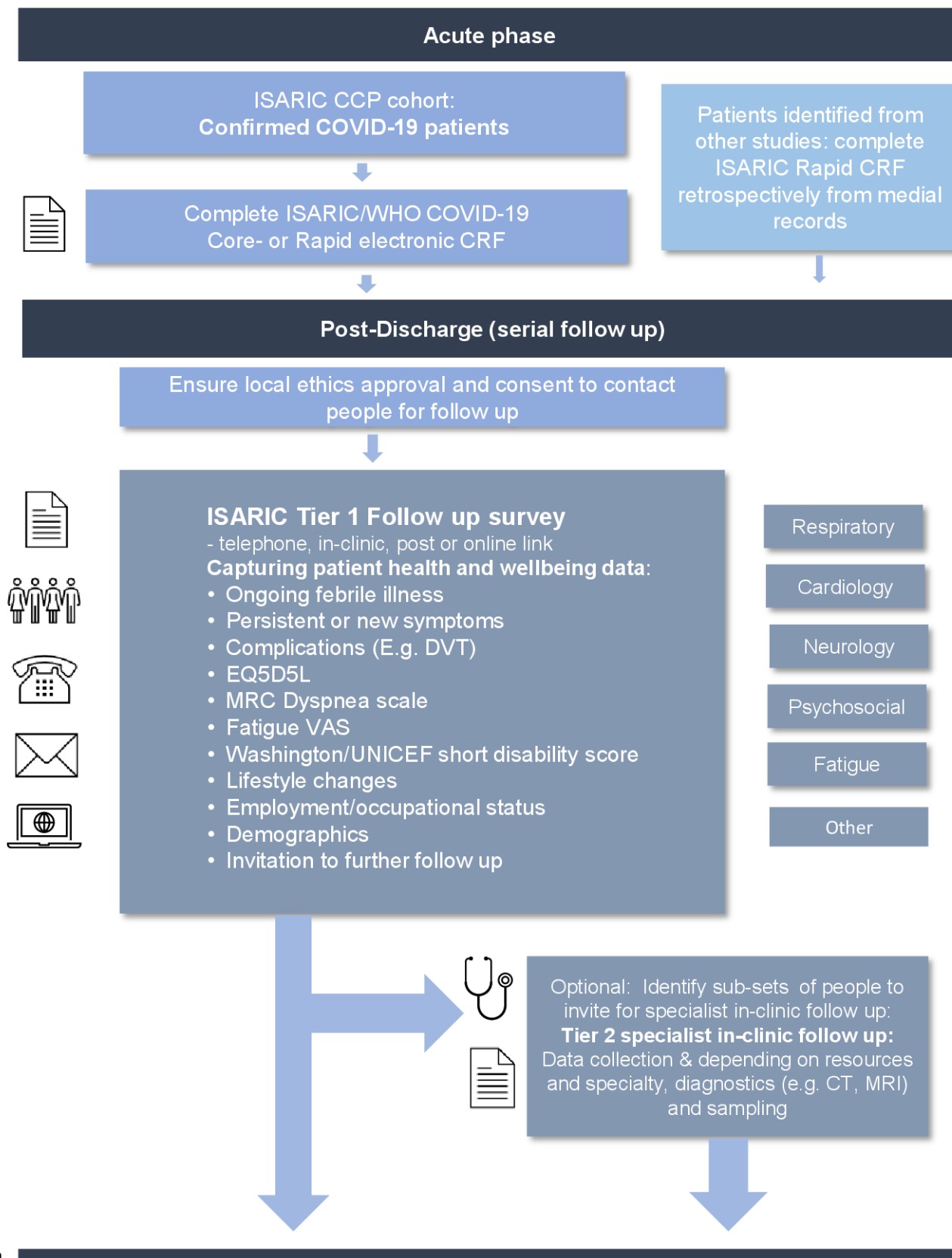

**Figure 1** ISARIC's adaptable COVID-19 follow-up protocol framework. CRF, case report form; CCP, Clinical Characterisation Protocol; DVT, deep vein thrombosis; ISARIC, International Severe Acute Respiratory and Emerging Infection Consortium; MRC, Medical Research Council.

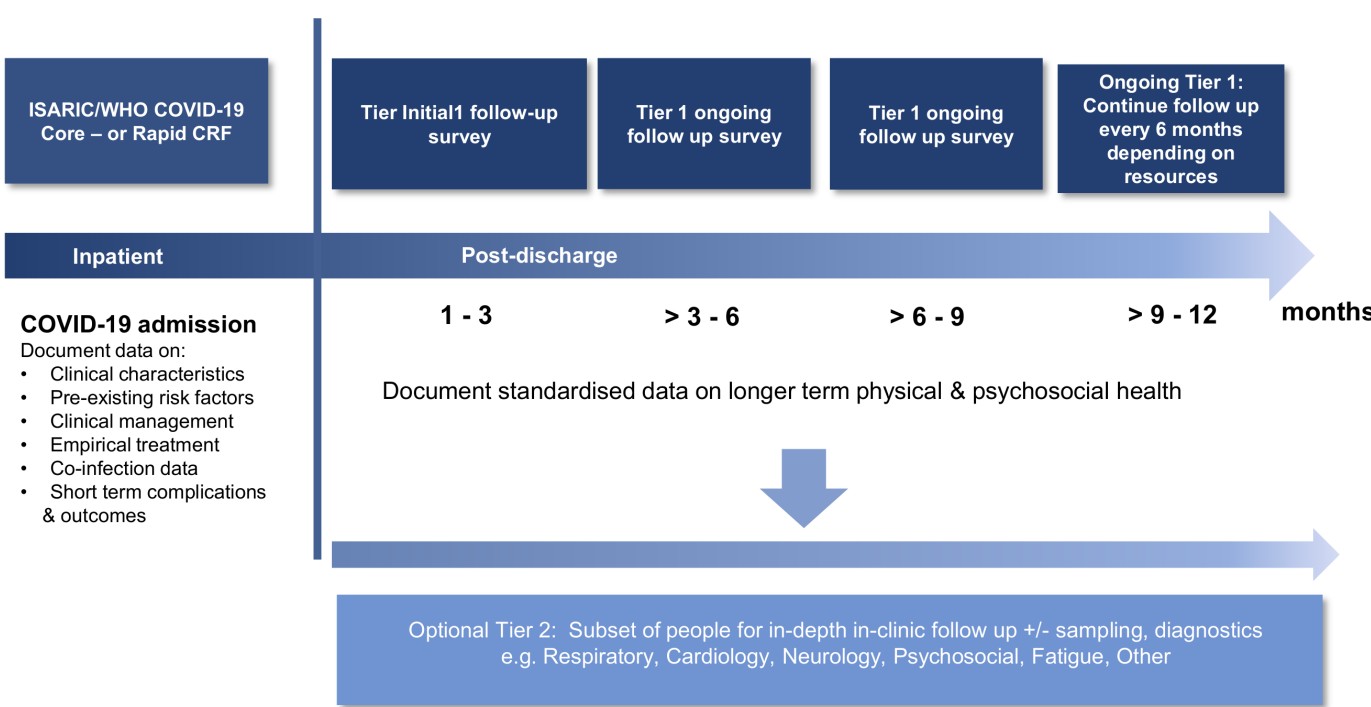

**Figure 2** Schematic overview of the follow-up data time frames. Abbreviations: CRF, case report form; ISARIC, International Severe Acute Respiratory and Emerging Infection Consortium.

clinical observational research response to any severe or potentially severe acute infection of public health interest, such as COVID-19. It is a standardised protocol for data and biological samples to be collected rapidly in a globally harmonised manner.[21 22] The CCP was activated in January 2020 to respond to COVID-19 worldwide. It is designed in a tiered approach and includes different levels of sampling schedules (acute phase and follow-up) that can be adapted depending on resources to be combined with patient data collection using the acute phase CRFs and the follow-up CRF.[18 19]

### Outcomes

The primary outcome of this study is to characterise physical and psychosocial consequences in patients postacute COVID-19. Secondary outcomes include estimating the risk of and risk factors for post-COVID-19 medical sequalae, psychosocial consequences and post-COVID-19 mortality.

### Data collection and entry

A standardised COVID-19 follow-up survey aimed for global settings was developed through a series of virtual working group meetings and email iterations. The survey was piloted on patients in four settings in three countries, and feedback was incorporated into the final form. The form collects data on a wide range of outcomes including hospital stay and all-cause and cause-specific mortality (after the initial index event), new or persistent symptoms and complications, for example, DVT, pulmonary embolism, recent febrile illness, new, persistent symptoms, EQ-5D-5L, MRC dyspnoea scale, UN/Washington disability score, lifestyle and

employment data. By standardising data collection, and providing open-access, adaptable tools focused on key data variables, it can optimise data quality and reduce the burden of research on staff while collecting the most relevant information to inform clinical care guidelines and public health.

This tier 1 follow-up survey is designed for patient self-assessment, via online link or paper form, or to be completed during in-clinic or telephone follow-up appointment in settings globally (figure 1).[18] The CRF is available open access on the ISARIC website (https://isaric.org/research/covid-19-clinical-research-resources/covid-19-long-term-follow-up-study/) and as an electronic form on the ISARIC hosted REDCap database.[18]

### Statistical analysis plan

Using the data, we will test for differences in outcomes across important demographic groups (age categories, sex, ethnicity, socioeconomic deprivation and comorbidities), specific exposures (severe COVID-19, critical care admission and ventilation) and initial clinical sequelae (complications on their index admission for COVID-19). We plan to use this platform to conduct timely analyses that coincide with public health or scientific need. Given these requirements, new questions we have not specified within this protocol may arise. Where this occurs, we will develop analysis plans prior to undertaking analyses, which will be made available on request. The data collected through the follow-up module will be linked with data on demographics, comorbidities, clinical characteristics, care and treatments collected using the ISARIC/WHO Core or RAPID COVID-19 CRF.[19]

Fields contained within the data collection forms will be combined, and if an area of interest is found, the maximal amount of data will be used to investigate this to maintain sample size and power. The plan below presents our guiding statistical framework. Analyses will be developed concurrently with data collection using statistical coding script. At appropriate time intervals, scripts will be run to produce analyses at these timepoints over the course of the project.

As COVID-19 is a new disease, there are no systematically collected long-term data to base formal sample size calculations on. Therefore, we intend to recruit as many patients as possible. Through the network established already, this is anticipated to be very large. Therefore, as a minimum calculation, to perform logistic regression, we will use at least 10 events for each variable included in the regression. For a regression containing 10 explanatory variables, 100 events would be included within the model for each variable. Assuming a sequelae rate of at least 20%, we would need at least 500 patients that should be adequately achieved. As the field is rapidly evolving, the analyses are likely to change.

Entered data will be summarised first by using simple summary statistics. Categorical data will be explored using frequencies and percentages, with differences in disease severity and treatment groups tested for using $\chi^2$ tests or Fisher's exact test where cell counts are under five. For continuous data, distribution will be established using histograms and density plots. Data that are normally distributed will be summarised using group mean averages and SD as a measure of central tendency. For nonparametric data, the median average will be used and presented alongside 25th and 75th centiles. Differences in normally distributed continuous data will be tested using Welch's two sample t-tests for two group data and analysis of variance for three or more groups. Mann-Whitney U test will be used to compare differences across two groups or Kruskall-Wallis tests for three or more groups, where data follow a non-parametric distribution.

Outcomes will be expressed in three ways: (1) binary event data (for presence or absence of outcome of interest), (2) change over time (for continuous or ordinal data) or as (3) time to event data (for patients with serial measurements). We will calculate changes over time across symptom and outcome variables and use these changes over time to compare the effect of treatments or exposures on these outcomes. Time to event data will be captured for those who complete serial assessment forms. These data will be presented as Kaplan-Meier plots and differences tested for using log-rank tests. Competing risks (including death) will be accounted for using censoring.

To address the main aim of characterising the middle to long-term impact of COVID-19 on physical and mental health, we will first provide simple summaries of incidence, and second, characterise which patients are at risk of developing these. To identify which patients are likely to develop persistent complications, functional impairment or reduced quality of life, we will use multilevel models to adjust for potential confounders. Level I fixed effects will include patient-level explanatory variables (ie, age and sex) and level II or III random effects will include site and country in the case where research questions require differences in country to be accounted for. Explanatory variables will be entered into models since clinical plausibility and final model selection guided by maximisation of the adjusted $R^2$ value and minimisation of the Akaike information criterion or Bayesian information criterion. For binary event data, multilevel logistic regression will be used, and estimates will be presented as ORs alongside the corresponding 95% CI. For continuous data, linear or generalised linear regression will be used, and estimates will be presented as model coefficients, with 95% CIs. Finally, time to event data will be presented as survival probability or HRs, with 95% CIs.

Statistical significance will be taken at the level of $p < 0.05$ a priori. Analyses will be conducted in secure R (R Foundation for Statistical Computing, Vienna, Austria) or STATA (StataCorp LLC, Texas, USA) environments.

Data-sharing sites who wish to use ISARIC's COVID-19 data management and hosting support can email ncov@isaric.org to gain access to a secure data capture and management system. All systems are free to use and supported by ISARIC data management specialists. Sites who submit data to ISARIC will sign a Terms of Submission, enabling use of the data in collaborative analysis by ISARIC partners. Data entered into the ISARIC database are stored on the servers administered by University of Oxford Medical Sciences Division Information Technology Services (MSD ITS). Data are stored in two active primary data storage locations (Data Centres) that host mirrored copies of data, one located near the MSD ITS offices at the John Radcliffe Hospital and the second at the Old Road Campus Research Building (sites A and B, respectively, both in Headington, Oxford, UK).

## Patient and public involvement

This protocol and the data collection survey has been informed by people living with long COVID-19. It has been piloted with a number of patients in different settings, and feedback was incorporated in the final version. This included suggestions on the data on symptoms collected and the way questions were asked as well as on the patient information.

## Ethics and dissemination

The protocol and survey are open access to enable low-resourced sites to join the study to facilitate global standardised, longitudinal data collection. The tools[18] are available in a range of languages.

Data from combined analysis will be disseminated through the ISARIC website and in open-access publications under group authorship. Ethical approval has been given by the Universidad de La Sabana's IRB, Colombia; Ghana Health Service Ethics Review Committee; at the Comitato Etico di Brescia, Italy; South-Eastern Norway Regional Health Authority, Norway; the Comitato

Etico per la sperimentazione Clinica delle Province di Verona e Rovigo, Italy; the Sechenov University Ethics Committee, Moscow, Russian Federation; and the University of Witwatersrand Human Research Ethics Committee (Medical), Johannesburg, South Africa. In the UK, for day – 28 follow-up as part of the UK CCP approved by the South Central - Oxford C Research Ethics Committee in England and the Scotland A Research Ethics Committee. New sites are welcome to join this collaborative study at any time. Sites interested in adopting the protocol as it is or in an adapted version are responsible for ensuring that local sponsorship and ethical approvals are in place as appropriate.

## Data statement

ISARIC stands by the principles of data sharing in public health emergencies. ISARIC supported studies will share quality data in a timely, valid and governed manner to inform public health policy and benefit patient care. The ISARIC hosted data platform enables rapid and harmonised operationalisation of data collection to a secure database.[23] Ownership and control of the data entered are retained by those who enter the data. Sites can contribute data to combined analysis on permission. Technical appendix, statistical code and datasets are available from https://isaric.org/document/covid-19-data-management-hosting/ or by contacting: ncov@isaric.org. The data contributors will share the outcome results with key stakeholders to inform public health response, policy development and implementation. To efficiently make findings from this project visible and accessible to a wide range of stakeholders as well as to networks of individuals and institutions, we will use social media, including the ISARIC and partner institution's websites. The consortium will manage a page dedicated to this project that will incorporate all the tools available. The webpage will provide updates on the progress of the project and component activities and events and information on research.

This follow-up study is developed as an open-access tool to be adopted as appropriate by any site interested in following up patients with COVID-19 over time to facilitate standardised data collection globally to enable combined analysis. New sites globally are invited to join the study at any time. The outcomes of this study will inform strategies to prevent risk of consequences; clinical management, rehabilitation and public health management needs to reduce morbidity; and improve outcomes. We invite hospitals and healthcare centres globally to collaborate and take part in the study.

## Author affiliations
[1]ISARIC Global Support Centre, Centre for Tropical Medicine and Global Heatlh, University of Oxford, Oxford, UK
[2]Infection and Global Health Division, School of Medicine, University of St Andrews, St Andrews, UK
[3]Department of Rehabilitation Medicine, NHS Lothian, Edinburgh, UK
[4]Department of Respiratory Medicine, Nottingham University Hospitals NHS Trust, Nottingham, UK
[5]Centro de Investigación Biomédica en Red – Enfermedades Respiratorias (CIBERES), Hospital Universitari Vall d'Hebron, Barcelona, Spain
[6]Research Department, CHU Nîmes, Université Nîmes-Montpellier, Nîmes, France
[7]School of Public Health, Kwame Nkrumah University of Science and Technology, Kumasi, Ghana
[8]Fundação Oswaldo Cruz, Rio de Janeiro, Brazil
[9]Department of Molecular and Clinical Cancer Medicine, Institute of Systems, Molecular and Integrative Biology, University of Liverpool, Liverpool, UK
[10]Clatterbridge Cancer Centre NHS Foundation Trust, Livepool, UK
[11]Department of Paediatrics, I M Sechenov First Moscow State Medical University, Moskva, Russia
[12]IInflammation, Repair and Development Section, National Heart and Lung Institute, Imperial College London Faculty of Medicine, London, UK
[13]Department of Microbiology, Oslo University Hospital, Oslo, Norway
[14]Institute of Clinical Medicine, University of Oslo, Oslo, Norway
[15]Department of Anesthesiology and Intensive Care, University Hospital of North Norway, Tromso, Norway
[16]Universidad de La Sabana, Chia, Colombia
[17]The University of Edinburgh Centre for Inflammation Research, Edinburgh, UK
[18]University of Glasgow, Glasgow, UK
[19]NIHR Health Protection Research Unit in Emerging and Zoonotic infections, Institute of Infection, Veterinary and Ecological Sciences, University of Liverpool, Liverpool, UK
[20]Tropical and Infectious Disease Unit, Liverpool University Hospitals NHS Foundation Trust, Liverpool, UK
[21]Centre for Medical Informatics, The University of Edinburgh, Edinburgh, UK
[22]Department of Infectious Diseases, Tropical and Microbiology, IRCCS Sacro Cuore Don Calabria Hospital, Negrar di Valpolicella, Italy
[23]Sustainable Health Systems, Freetown, Sierra Leone
[24]Dorothy Springer Trust, Freetown, Sierra Leone
[25]Sunnybrook Health Sciences Institute, Sunnybrook Research Institute, Toronto, Ontario, Canada
[26]Institute of Cardiovascular and Medical Sciences, BHF Glasgow Cardiovascular Research Centre, University of Glasgow, Glasgow, UK
[27]Emergency Department, Queen Elizabeth University Hospital, Glasgow, UK
[28]NHS Greater Glasgow and Clyde, Glasgow, UK
[29]Institute of Health and Wellbeing, University of Glasgow, Glasgow, UK
[30]Department of Critical Care Medicine, Ziauddin University, Karachi, Pakistan
[31]Department of Infectious Diseases, Institute of Clinical Medicine, Oslo University Hospital, Oslo, Norway
[32]National Institute of Health Research (NIHR) Health Protection research Unit in Emerging and Zoonotic Infections, University of Liverpool, Liverpool, UK
[33]Institute of Infection and Global Health, Faculty of Health and Life Sciences, University of Liverpool, Liverpool, UK
[34]Nuffield Department of Medicine, ISARIC Global Support Centre, Centre for Tropical Medicine and Global Health, University of Oxford, Oxford, UK
[35]London School of Hygiene & Tropical Medicine, London, UK
[36]Usher Institute, The University of Edinburgh, Edinburgh, UK
[37]Centre for Medical Informatics, Usher Institute, The University of Edinburgh, Edinburgh, UK
[38]Division of Genetics and Genomics, The University of Edinburgh The Roslin Institute, Roslin, UK
[39]Health Protection Research Unit In Emerging and Zoonotic Infections, Institute of Infection, Veterinary and Ecological Sciences, University of Liverpool, Liverpool, UK
[40]University of Liverpool, Alder Hey Children's NHS Foundation Trust, Liverpool, UK
[41]MRC, University of Glasgow Centre for Virus Research, Glasgow, UK

**Acknowledgements** We would like to thank the International Severe Acute Respiratory and Emerging Infection Consortium (ISARIC) global clinical characterisation group, ISARIC4C and all the clinicians, nurses and researchers contributing COVID-19 clinical patient data, which will be linked with the follow up data, and all the patients that have consented to be followed up. We would like to acknowledge WHO, whose working groups contributed to development of the clinical characterisation protocol and associated tools. Moreover, Sarah Moore, Romans Matulevics, James Lee, Laura Merson, Peter Bannister, Katherine Maskell,

Eli Harriss, Liliana Resende, Anneli Sandström and Tova Strong for administrative, graphic design and dissemination support.

**Contributors** JTS and LS lead on the development of the follow-up protocol and tools (survey, consent form and patient information sheet) in collaboration with members of the ISARIC COVID-19 follow-up working group (JR, JA, FB, CP, DM, JCH, ABK, CDR, TMD, AMD-R, KH, IRB, AB, MH, RF, SL, LFR, AH, CD, HH, NE, LT, CR, MC and WSL) and external specialists (EJ, DJL, CB, JM, NL, MGS, JKB, EMH, AD and GC) through a series of meetings and email iterations. AH, LT, ABK, JR and CP piloted the case report form in the clinic, before it was finalised by the members of the working group. TMD, LFR, LS, JTS, AD and EMH developed the statistical analysis plan. DRP, HH and CD managed the database set up on REDCap and individual site set up. LS, MC, TMD and JTS lead on drafting of the protocol and manuscript, with contributions from JR, JA, FB, CP, DM, JCH, ABK, CDR, TMD, AMD-R, KH, IRB, AB, MH, RF, SL, LFR, AH, CD, HH, NE, LT, CR, MC, WSL, EJ, DJL, CB, JM, NL, MGS, JKB, EMH, AD, GC. All authors reviewed and approved the final manuscript.

**Funding** This work was supported by the Department for International Development and Wellcome (215091/Z/18/Z) and the Bill & Melinda Gates Foundation (OPP1209135). CP would like to acknowledge the support of the Liverpool Experimental Cancer Medicine Centre (Grant Reference: C18616/A25153) and The Clatterbridge Cancer Centre Charity. CB acknowledges the support from British Heart Foundation RE/18/6134217. LS would like to acknowledge the support of PREPARE funded by the European Commission's FP7 Programme grant number 602525.

**Competing interests** None declared.

**Patient and public involvement** Patients and/or the public were involved in the design, or conduct, or reporting, or dissemination plans of this research. Refer to the Methods section for further details.

**Patient consent for publication** Not required.

**Ethics approval** Ethical approval has been given by the Universidad de La Sabana's IRB (US-MED-504), Colombia; Ghana Health Service Ethics Review Committee; the Comitato Etico di Brescia, Italy; and South-Eastern Norway Regional Health Authority, Norway (Ref 106624). In the UK, for day 28 follow up as part of the UK CCP approved by the South Central – Oxford C Research Ethics Committee in England (Ref 13/SC/0149) and the Scotland A Research Ethics Committee (Ref 20/SS/0028). The protocol is under review at the Comitato Etico per la sperimentazione Clinica delle Province di Verona e Rovigo, Italy, the Sechenov University Ethics Committee, Moscow, Russian Federation and the University of Witwatersrand Human Research Ethics Committee (Medical), Johannesburg, South Africa and is being submitted to the National Ethics Committee (CONEP), Rio de Janeiro, Brazil, and in Freetown, Sierra Leone. Sites interested in adopting the protocol as it is or in an adapted version are responsible for ensuring that local sponsorship and ethical approvals in place as appropriate.

**Provenance and peer review** Not commissioned; externally peer reviewed.

**ORCID iDs**
Louise Sigfrid http://orcid.org/0000-0003-2764-1177
Jordi Rello http://orcid.org/0000-0003-0676-6210
Daniel Munblit http://orcid.org/0000-0001-9652-6856
Anders Benjamin Kildal http://orcid.org/0000-0002-1319-6511
Joanne McPeake http://orcid.org/0000-0001-8206-6801

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
