## [Reviewer comments · BMJ Open]

ARTICLE DETAILS

TITLE (PROVISIONAL)	What is the recovery rate and risk of long-term consequences following a diagnosis of COVID-19? – a harmonised, global longitudinal observational study protocol
AUTHORS	Sigfrid, Louise; Cevik, Muge; Jesudason, Edwin; Lim, Wei Shen; Rello, Jordi; Amuasi, John; Bozza, Fernando; Palmieri, Carlo; Munblit, Daniel; Holter, Jan; Kildal, Anders Benjamin; Reyes, Luis; Russell, Clark; Ho, Antonia; Turtle, Lance; Drake, Thomas; Beltrame, Anna; Hann, Katrina; Bangura, Ibrahim; Fowler, Robert; Lakoh, Sulaiman; Berry, Colin; Lowe, David; McPeake, Joanne; Hashmi, Madiha; Dyrhol-Riise, Anne Margarita; Donohue, Chloe; Plotkin, Daniel; Hardwick, Hayley; Elkheir, Natalie; Lone, Nazir; Docherty, Annemarie; Harrison, Ewen; Baille, J; Carson, Gail; Semple, Malcolm G; Scott, Janet

VERSION 1 – REVIEW

REVIEWER	Lisa Sheehy Bruyère Research Institute, Canada
REVIEW RETURNED	07-Oct-2020

GENERAL COMMENTS	Summary This multicentre, international cohort study will follow a large number of patients admitted to hospital for treatment for COVID-19, for up to a year. Demographics, characteristics of patients' COVID-19 signs and symptoms, treatment and progression of recovery will be tracked. Analyses include investigation of the risk factors for COVID-19 and the consequences of serious illness (requiring hospitalization). This is an important study and the results will provide vital insight into the impact of this novel coronavirus. General Comments Overall, I found this manuscript very difficult to follow. It was not easy to understand the procedure and Figure 2 was somewhat confusing. The manuscript should also be reviewed to ensure that the English is clear and unambiguous, at a level appropriate for scientific reporting. Specific Comments Title
--

- The title should include the term 'protocol'.

Abstract

- The Aims should be at the end of the Introduction.
- Add that the study will follow patients hospitalized for COVID-19, and that these patients have already had a CRF filled out (and what types of data are on the CRF).
- It would be clearer to state the actual times of follow-up. From Figure 1, this would be 28 days, 3 months, 6 months and 12 months post-discharge.
- Be clear about what the assessments are at the different times, and for each tier.
- Briefly mention the analyses to be done for each aim.
- For the subset of patients that have sampling, what aim is this related to? What types of sampling?

Article Summary

- Write out acronyms.
- Mention the languages that the report forms are available in.

Introduction

- I could not find reference #7; the reference might need updating.
- In the last paragraph of the introduction, please state clearly all Aims, as in the abstract.

Methods and Analyses

Study Design

- Add the term "cohort" to your study design sentence.
- Declaration of conformity to research ethics and review by a research ethics board.

Population and Setting

- Add more detail about the ISARIC/WHO protocol at the beginning of this paragraph. Explain what the Core and RAPID Case reports are (include as appendices or online extras if appropriate). Only after I went to the website did I understand what this was.
- How will the CRF data be linked with the follow-up data?
- When and by whom are the CRFs administered? Are they done at discharge from acute care? Or earlier on in their hospital stay? How can the CRFs be completed retrospectively? What does this mean? Will the patient have already left the hospital? This is confusing in a "prospective" study.

	 • What is your target N? How was the sample size determined? Inclusion of Vulnerable Participants  • Since your inclusion criteria exclude children, this statement can be deleted. Outcomes and Procedure  • This should be the next sub-heading, combining the current headings of “Serial follow up”, “Sub-studies”, “Biological samples” and “Outcomes”. Outline step-by-step what the participants will do, with all testing measures noted and the times that each will be done. Clearly define “Tier 1” and “Tier 2” and what each entails. • Who administers the testing? Local research assistants/administrators, or a central team of researchers? Even self-completion forms need to be sent by somebody. Is it all done on the phone or over the internet? Actually, it seems like some of the data need to be reported by a healthcare professional. • Provide a complete list of data to be acquired. This might be in the form of a table for an online extra. • What languages will it be available in? • How will follow-up occur for patients who die? • Two grammar errors which are important to fix:  page 8, line 55 “... 3 to 6 months interval, with ...” should be “... 3 to 6 month intervals, with ...”. Page 9, line 2 “... new persistent ...” should be “ ... new and/or persistent ...” • State when exactly the self-assessments will be done. Figure 1 shows at 3, 6 and 12 months for Tier 1 and 6 and 12 months for Tier 2. • What is the expected compliance rate with the follow-up testing? • For Tier 2, there seems to be leeway given to add modules in the future, but give an example of what the research team might be looking for. • “Outcomes” – this information should be put at the end of the introduction, as the aims of the study. • As it is written here, page 10 line 2, this is the first I have heard about tracking re-exposure and re-infection. Data Collection and Entry  • I would place this paragraph under Population and Setting, when you talk about the CRFs. Statistical Analysis Plan
--	--

- Relate the analyses back to the specific Aims.
- How will the different information contained on the Core and RAPID CRFs affect the analysis? i.e. there will be different N's for different demographic factors and outcomes.
- When will the analysis take place? If centres are being invited to join this project at any time, when will interim and final analyses take place?

Data Sharing

- Where is the ISARIC database located? Is it on the cloud or on a server?

Patient and Public Involvement

- I would put this paragraph earlier, when you are talking about the creation of "the survey". (What "survey" is this, the CRF, or the other outcomes to be done in Tier 1?).

Ethics and Dissemination

- This section should be removed.
- The first paragraph should be moved to the introduction. This may introduce repetition which can be edited out.
- Parts of the second paragraph should be moved to Population and Setting, and duplication removed. Of note, some things are mentioned here that were not mentioned before (ex. Re-admissions). In the list of consequences, it should read new and/or persistent symptoms. EQ-5D-5L etc. are not "consequences", per se. These sentences and the first part of the third paragraph should be incorporated into the Outcomes and Procedure section.
- Parts of the 3rd paragraph should be moved into a new "Discussion" section. This should detail the expected outcomes and clinical importance of the study. Limitations should be included here too.

Data Statement

OK

Discussion

- This section should be added.
- Limitations should be included (ex. Loss to follow-up and bias; ability to track deaths)

Figure 1

- In the first column it says "COVID-19 onset". Is this true? If so, it should be clear in the text of the manuscript. If it is true, it will be hard to track things like treatment and outcomes of acute care.

	 • In the first column is says Medical records. This is the first instance of mentioning obtaining data from medical records. • Column 2 mentions sampling. What type of sampling? This is not mentioned with respect to Tier 1 in the manuscript. • The arrow points to the right. Will there be more assessments beyond that done at 12 months? This is unclear in the manuscript and this diagram tells me that assessments might continue. • The term COVID-19 is written inconsistently in Figure 1 and Figure 2. Figure 2  • I found this very confusing. I am uncertain why there are 2 parallel pathways. Don't all patients have to be both confirmed as having COVID-19 and being in the ISARIC cohort? This would make 1 pathway. • "Confirmed COVID-19 patients" comes after "Recovered COVID-19 patients". What does this mean? Doesn't a patient have to have confirmed COVID-19 before recovering from it? • Where is Tier 2? Does everyone enter Tier 2? Maybe this should split off after Tier 1? • The open bubbles with "Immunology/Virology/Microbiomes" etc. – these have not been mentioned in the manuscript. If there is too much detail for the manuscript, a list of possible Tier 2 studies should be added as an online extra. • Visually, it looks like the Integrated Data Platform goes to the open bubbles. Either put it below the other bubbles or add arrows. The flow chart might be best if it was laid out in portrait format.
--	--

REVIEWER	Ali Gholamrezanezhad Keck School of Medicine, University of Southern California (USC) Los Angeles, California, USA
REVIEW RETURNED	18-Oct-2020

GENERAL COMMENTS	I had the pleasure to review the manuscript entitled "What is the recovery rate and risk of long-term consequences following a diagnosis of COVID-19? - A harmonised, global longitudinal observational study ". The topic of long-term pulmonary consequences of COVID-19 is a very important area that needs extensive research and I really appreciate the authors effort on this area. My main concern is the fact that based on the available data from similar coronavirus infections, such as SARS and MERS, a significant number of patients with these diseases may have subclinical sequel. These subclinical scars of the disease can be easily unveiled by CT and pulmonary function test. Please refer to the publication below and consider discussing it in your discussions:
--

	- Salehi S, Reddy S, Gholamrezanezhad A. Long-term Pulmonary Consequences of Coronavirus Disease 2019 (COVID-19): What We Know and What to Expect. J Thorac Imaging. 2020 Jul;35(4):W87-W89. doi: 10.1097/RTI.0000000000000534. PMID: 32404798. - Shaw B, Daskareh M, Gholamrezanezhad A. The lingering manifestations of COVID-19 during and after convalescence: update on long-term pulmonary consequences of coronavirus disease 2019 (COVID-19). Radiol Med. 2020 Oct 1:1–7. doi: 10.1007/s11547-020-01295-8. Epub ahead of print. PMID: 33006087; PMCID: PMC7529085. I mean I am not sure about the sensitivity of the study in unmasking long-term sequela of the disease by ignoring imaging (such as CT) and pulmonary function testing. What if the patients has asymptomatic scar? You consider them as "no long-term consequence"? These patients are clearly affected by the disease and their long-term prognosis is not similar to those without scar in imaging. Also, as a reviewer, I would like to know what risk factors will be collected in this study to evaluate the risk of long-term consequence. Please name all the variables that will be collected in this prospective study. - Overall, the methods are not described and referenced adequately and in sufficient detail for reproducibility. Experimental and control groups, including inclusion and exclusion criteria, are not clearly identified and described. How measurements were made and what calculations were performed are not appropriately explained. - What is your plan to assess the effect of the type of treatment on the risk of long-term consequences? - Please discuss potential study limitations.
--	--

REVIEWER	Valeria Cento Università degli Studi di Milano, Milano (Italy)
REVIEW RETURNED	18-Oct-2020

GENERAL COMMENTS	The project presented here has some elements of interest. It certainly responds to a pressing need, also clinical, but above all social, to better understand the impact of this pandemic on the psychological, clinical and societal fabric of the affected individuals, as well as their families. I am not sure, however, that a "project design" would fit a peer-reviewed scientific publication.
--

VERSION 1 – AUTHOR RESPONSE

Reviewer(s)' Comments to Author:

Reviewer: 1
Reviewer Name
Lisa Sheehy

Institution and Country
Bruyère Research Institute, Canada

Please state any competing interests or state 'None declared':
None declared

Please leave your comments for the authors below
See attached file.

Reviewer: 2
Reviewer Name
Ali Gholamrezanezhad

Institution and Country
Keck School of Medicine, University of Southern California (USC)
Los Angeles, California, USA

Please state any competing interests or state 'None declared':
None declared

Please leave your comments for the authors below

I had the pleasure to review the manuscript entitled "What is the recovery rate and risk of long-term consequences following a diagnosis of COVID-19? - A harmonised, global longitudinal observational study ". The topic of long-term pulmonary consequences of COVID-19 is a very important area that needs extensive research and I really appreciate the authors effort on this area.

My main concern is the fact that based on the available data from similar coronavirus infections, such as SARS and MERS, a significant number of patients with these diseases may have subclinical sequel. These subclinical scars of the disease can be easily unveiled by CT and pulmonary function test. Please refer to the publication below and consider discussing it in your discussions: - Salehi S, Reddy S, Gholamrezanezhad A. Long-term Pulmonary Consequences of Coronavirus Disease 2019 (COVID-19): What We Know and What to Expect. J Thorac Imaging. 2020 Jul;35(4):W87-W89. doi: 10.1097/RTI.0000000000000534. PMID: 32404798.

- Shaw B, Daskareh M, Gholamrezanezhad A. The lingering manifestations of COVID-19 during and after convalescence: update on long-term pulmonary consequences of coronavirus disease 2019 (COVID-19). Radiol Med. 2020 Oct 1:1–7. doi: 10.1007/s11547-020-01295-8. Epub ahead of print. PMID: 33006087; PMCID: PMC7529085.

A: thank you for reviewing the protocol submission and for your helpful comments and citations. We have incorporated this - line 211. " Additionally, serial CT scans post discharge after SARS-CoV showed a gradual healing of pulmonary injury, with pulmonary consequences lasting more than 6 months post discharge. (15)"

I mean I am not sure about the sensitivity of the study in unmasking long-term sequela of the disease by ignoring imaging (such as CT) and pulmonary function testing. What if the patients has asymptomatic scar? You consider them as "no long-term consequence"? These patients are clearly affected by the disease and their long-term prognosis is not similar to those without scar in imaging.

A: This study is based on the emerging reports of people reporting persistent, or new symptoms post-acute Covid-19 and the data collected on physical and psychosocial symptoms and complications informed by specialists seeing people in post-acute Covid clinics and by Long Covid patient support groups. The aim is at a tier 1 level document data to characterise post-acute Covid-19 so called 'Long

Covid' to assess the breadth and prevalence of persistent and new complications experienced and identify people with specific symptoms to different more in dept follow up, which could include CT in patients presenting with fatigue, breathlessness etc.

The Tier 1 survey is designed to identify these patients from a wide population. That is the Tiered approach for wide uptake during a pandemic using Tier 1, to find those that are appropriate to follow up with diagnostics. It would not be economically feasible or appropriate to subject everyone to CT and other diagnostic methods. We have updated the text to clarify this approach and the scientific rationale more explicitly in the methods section.

Also, as a reviewer, I would like to know what risk factors will be collected in this study to evaluate the risk of long-term consequence. Please name all the variables that will be collected in this prospective study.

A: Thanks, we decided not to submit the survey with the article, instead refer people to the website to ensure they access the latest version in case there are future iterations. However, happy to submit it as an appendix for ease of access and to guide the reader.

- Overall, the methods are not described and referenced adequately and in sufficient detail for reproducibility. Experimental and control groups, including inclusion and exclusion criteria, are not clearly identified and described. How measurements were made and what calculations were performed are not appropriately explained.

A: this is an observational study set up for the timely follow up of people already included in the acute phase of the study set up in response to the reports of 'long covid' to rapidly forwarding the knowledge into the longer term outcomes from Covid-19. Observational studies are the corner stone for clinical research responses to emerging epidemics, that can be implemented in any resourced setting and often does not require lengthy ethical approval processes to characterise emerging infections in a timely manner to inform clinical management, public health interventions and interventional study designs. We agree there are also need for case control and interventional studies for long covid. This study can help inform these study designs.

- What is your plan to assess the effect of the type of treatment on the risk of long-term consequences? Please see the answer above. We do not attempt to assess treatment affect, but focus on descriptive data, e.g. scoping analysis to review association with clinical management including basic empirical treatment, and prolonged recovery/persistence of specific symptoms.

- Please discuss potential study limitations.

A; We have followed the authors guidelines for protocol submissions and updated the study limitations in the Strength and limitations section at the beginning of the protocol. The BMJ Open protocol guidelines does not include a discussion, instead a brief bullet point strength and limitation section at the beginning.

Reviewer: 3

Reviewer Name

Valeria Cento

Institution and Country

Università degli Studi di Milano, Milano (Italy)

Please state any competing interests or state 'None declared':

None declared

Please leave your comments for the authors below

The project presented here has some elements of interest. It certainly responds to a pressing need, also clinical, but above all social, to better understand the impact of this pandemic on the

psychological, clinical and societal fabric of the affected individuals, as well as their families. I am not sure, however, that a “project design” would fit a peer-reviewed scientific publication.

A: Thank you for reviewing the protocol submission. This is a protocol submitted under the BMJ Open protocol submission category, following these author guidelines. This is an open access study which we invite healthcare sites to join and collaborate on at any time, to help forward knowledge into Covid-19 and outcomes overtime. Hence, we are keen for wide dissemination to facilitate study uptake, to include people with different risk factors and populations globally to generate sufficient numbers for robust and internationally comparative analysis. All combined analysis will be published under a group authorship recognising all site contributors in a truly collaborative way.

FORMATTING AMENDMENTS (if any)

Required amendments will be listed here; please include these changes in your revised version:

A: Thank you for your comments. The study protocol submitted will address these pressing questions. This is an open access observational study protocol which is provided as a collaborative study and template for any sites globally to adapt and use for independent studies, or to contribute data to the combined analysis. The aim is to make the protocol widely accessible, to sites who may not have the resources to develop protocol and data collection forms. By providing open accessible templates can facilitated standardisation of data collected at multiple sites worldwide, and generate sufficient, robust data across clinical spectra and demographics, different risk factors to enable statistically valid results.

This is article is submitted under the BMJ Open Research protocol submission category.

Reviewer one’s appendix :

Summary

This multicentre, international cohort study will follow a large number of patients admitted to hospital for treatment for COVID-19, for up to a year. Demographics, characteristics of patients’ COVID-19 signs and symptoms, treatment and progression of recovery will be tracked. Analyses include investigation of the risk factors for COVID-19 and the consequences of serious illness (requiring hospitalization). This is an important study and the results will provide vital insight into the impact of this novel coronavirus.

General Comments

- Overall, I found this manuscript very difficult to follow. It was not easy to understand the procedure and Figure 2 was somewhat confusing. The manuscript should also be reviewed to ensure that the English is clear and unambiguous, at a level appropriate for scientific reporting.

A: Thank you for reviewing the protocol submission and providing your very detailed and useful comments. The language has been reviewed by an information specialist and updated where appropriate throughout. Figure 1 and 2 has had their content updated to clarify the process.

The text has been revised to clarify the set up and the link with the existing cohort of clinical data documented during the acute admission, including the forms used for this. Moreover, how this study is an extension of this acute phase study.

Specific Comments

Title

- The title should include the term ‘protocol’.

A: Good point, thanks. It has been added.

Abstract

- The Aims should be at the end of the Introduction.

A: Thanks, for highlighting this, the aims have been moved to the end of the introduction.

- Add that the study will follow patients hospitalized for COVID-19, and that these patients have already had a CRF filled out (and what types of data are on the CRF).

A: Thanks, we have clarified the methodology and included a description of the link with the acute phase study, and removed the timeframes from the abstract. They are described in the paper and the updated infographics.

- It would be clearer to state the actual times of follow-up. From Figure 1, this would be 28 days, 3 months, 6 months and 12 months post-discharge.

A: Thanks, we have clarified the methodology, the figures and the follow up time frames in Fig. 2.

Be clear about what the assessments are at the different times, and for each tier.

A: Thanks, we have clarified the methodology and included a description of the link with the acute phase study, and removed the timeframes from the abstract. They are described in the paper and the updated Fig. 1.

- Briefly mention the analyses to be done for each aim.

A: We have added a short sentence to describe the overarching strategy of the analyses.

Unfortunately, due to the multiple outcome measures and groups of patients, it is not practical to outline the analyses for each aim in just the study abstract.

- For the subset of patients that have sampling, what aim is this related to? What types of sampling?

A: We have clarified that the study provides data collection forms, and that these can be used for other optional studies for more in depth follow up, including e.g. sampling and diagnostics (MRI etc.). The tier 1 survey is set up so that it can identify people with set symptoms indicating e.g. cardiopulmonary, neurological or psychosocial sequelae to invite back for in clinic follow up (Tier 2) by sites with resources to do so. To clarify that this protocol is set up for the Tier 1 level, the higher tiers are optional additions for those that have resources to set these up and invite people identified at Tier 1 back.

Article Summary

- Write out acronyms.

A: Updated throughout

- Mention the languages that the report forms are available in.

A: The survey forms are currently available in Italian, Portuguese, Russian and Spanish. We will provide translations (of the paper version and online link) in additional languages upon request. We have clarified this in the protocol.

Introduction

- I could not find reference #7; the reference might need updating.

A: Thanks for highlighting this, reference 7 is no longer available from the source, it has been removed. I have therefore removed this sentence and citation from the protocol manuscript.

- In the last paragraph of the introduction, please state clearly all Aims, as in the abstract.

A: Updated.

Methods and Analyses

Study Design

- Add the term "cohort" to your study design sentence.

A: Updated.

- Declaration of conformity to research ethics and review by a research ethics board.

A: This has been added in the study design section. The sites that have ethical approval or are awaiting a decision are listed at the end.

Population and Setting

• Add more detail about the ISARIC/WHO protocol at the beginning of this paragraph. Explain what the Core and RAPID Case reports are (include as appendices or online extras if appropriate). Only after I went to the website did I understand what this was.

- How will the CRF data be linked with the follow-up data?

- When and by whom are the CRFs administered? Are they done at discharge from acute care?

- Or earlier on in their hospital stay? How can the CRFs be completed retrospectively? What does this mean? Will the patient have already left the hospital? This is confusing in a “prospective” study.

A: the text has been reviewed and updated to address the points above, to better clarify how this study builds on the current cohort of patients admitted to hospitals, that already have data documented during the acute phase using the Core or rapid forms.

And that we also welcome new sites to join the collaboration, in which case they can complete these acute forms from medical records.

This has been clarified throughout and in Fig. 2.

- What is your target N? How was the sample size determined?
- As COVID-19 is a new condition, there are no systematically collected long-term data to base formal sample size calculations upon. Therefore, the objective is to recruit the maximum number of patients possible. Through this network, we anticipate the sample size will be very large. We have added a sample size explanation. It is difficult to prespecify sample size for specific analyses given that long-term sequelae are not known.

Inclusion of Vulnerable Participants

- Since your inclusion criteria exclude children, this statement can be deleted.

A: The statement has been edited to avoid repetition, and further specified to state that the forms are developed to include additional vulnerable participants (elderly and those who are immunosuppressed) added. We want to highlight that we are in the process to develop aligned surveys to enable follow up of children and young people.

Outcomes and Procedure

- This should be the next sub-heading, combining the current headings of “Serial follow up”, “Substudies”

A: the heading Outcomes and Procedures has been added.

- “Biological samples” and “Outcomes”. Outline step-by-step what the participants will do, with all testing measures noted and the times that each will be done. Clearly define “Tier 1” and “Tier 2” and what each entails.

A: The explanation in regards to biological samples has been clarified (Please see previous answers). The description on what data will be documented is evident from the surveys. We have updated the links to the surveys. We prefer to refer people to download the surveys from a website to ensure they access the latest version, in case there are later revisions. Therefore we have not supplied them as a supplemental file.

- Who administers the testing? Local research assistants/administrators, or a central team of researchers? Even self-completion forms need to be sent by somebody. Is it all done on the phone or over the internet? Actually, it seems like some of the data need to be reported by a healthcare professional.

A; the description of the different administration avenues have been updated and we have added that sites are responsible for contacting patients, and ensuring local ethical regulations are in place.

ISARIC provides templates and a collaborative platform to facilitate uptake and implementation of epidemic research in LMIC to HICs and a shared database and data sharing agreements to facilitate timely combined analysis.

- Provide a complete list of data to be acquired. This might be in the form of a table for an online extra.

A: The Tier 1 Survey has been cited with the online links to access the survey and provided as a supplemental file.

- What languages will it be available in?

A: the survey is currently available on the ISARIC website in Italian, Portuguese, Russian and Spanish. We will translate it into any languages required on request.

- How will follow-up occur for patients who die?

A: Before contacting patient for follow up post-discharge, the study investigator will check that the patient is not recorded as deceased in the CRFs documented during the acute phase, or in the

medical records, as well as that they have consented to be contacted for follow up. This check will be done ahead of each serial follow up.

- Two grammar errors which are important to fix: page 8, line 55 "... 3 to 6 months interval, with ..." should be "... 3 to 6 month intervals, with ...".

A: Thanks, updated

- Page 9, line 2 "... new persistent ..." should be "... new and/or persistent ..."

A: Well spotted, thanks, updated

- state when exactly the self-assessments will be done. Figure 1 shows at 3, 6 and 12 months for Tier 1 and 6 and 12 months for Tier 2.

A: The timeframe has been clarified (pls see Fig.2)

- What is the expected compliance rate with the follow-up testing?

A: please see previous answers about sampling – this is optional/an add on for sites with resources to do this or who are already planning sampling studies. They may use , or adapt and adopt the Tier 1 surveys for their studies. They are open access.

- For Tier 2, there seems to be leeway given to add modules in the future, but give an example of what the research team might be looking for.

A: Please see previous answers. The text has been revised to clarify that Tier 2 is optional add on. We are not developing Tier 2 modules, as it requires different specialties to develop specific modules focusing on different symptomatology. Tier 1 can be used to identify patients to invite for different specialty follow ups.

- "Outcomes" – this information should be put at the end of the introduction, as the aims of the study.

A: This has been updated.

- As it is written here, page 10 line 2, this is the first I have heard about tracking re-exposure and re-infection.

- A: This has been updated.

Data Collection and Entry

- I would place this paragraph under Population and Setting, when you talk about the CRFs.

Statistical Analysis Plan

- Relate the analyses back to the specific Aims.
- We have clarified this in the statistical analyses.

- How will the different information contained on the Core and RAPID CRFs affect the analysis?

- This will not alter the analyses, we will combine this information together and use the essential variables contained across these.

- i.e. there will be different N's for different demographic factors and outcomes.

- We have added some information on number of patients we require to use modelling approaches.

- When will the analysis take place? If centres are being invited to join this project at any time, when will interim and final analyses take place?

- Analyses will be undertaken concurrently with data collection using R scripts, this allows real-time analysis and reduces delays.

Data Sharing

- Where is the ISARIC database located? Is it on the cloud or on a server?

• Data entered into the ISARIC database are stored on the servers administered by University of Oxford Medical Sciences Division Information Technology Services (MSD ITS). Data are stored in two active primary data storage locations (Data Centres) which host mirrored copies of data, one located near the MSD ITS offices at the John Radcliffe Hospital and the second at the Old Road Campus Research Building (sites A and B respectively, both in Headington, Oxford, UK).

Patient and Public Involvement

- I would put this paragraph earlier, when you are talking about the creation of “the survey”. (What “survey” is this, the CRF, or the other outcomes to be done in Tier 1?).

A: The text has been edited to make it clearer, and a more logical flow.

Ethics and Dissemination

- This section should be removed.

A: We followed BMJ Opens author guidelines for protocol which does not include a discussion, but ethics and dissemination after methods, no discussion or conclusion.

<https://bmjopen.bmj.com/pages/authors/#protocol> Section therefore not removed.

- The first paragraph should be moved to the introduction. This may introduce repetition which can be edited out.

A; This section has been shortened as discussed and a paragraph about one of the main aim of providing research templates added. „This study is open access for any sites globally to join the collaboration to characterise COVID-19 in different populations. Providing research tools that can be implemented for free in low resourced settings can increase equity in implementation of and inclusion in clinical research studies.“

- Parts of the second paragraph should be moved to Population and Setting, and duplication removed. Of note, some things are mentioned here that were not mentioned before (ex. Readmissions).

A: see above, the ethical aspects has been highlighted and some sentences removed.

- In the list of consequences, it should read new and/or persistent symptoms. EQ-5D-5L etc. are not “consequences”, per se. These sentences and the first part of the third paragraph should be incorporated into the Outcomes and Procedure section.

A: Thanks this has been updated.

- Parts of the 3rd paragraph should be moved into a new “Discussion” section. This should detail the expected outcomes and clinical importance of the study. Limitations should be included here too.

A: We followed BMJ Opens author guidelines for protocols which does not include a discussion, but ethics and dissemination after methods, no discussion or conclusion.

<https://bmjopen.bmj.com/pages/authors/#protocol>

Data Statement

- OK

Discussion

- This section should be added.

A: We followed BMJ Opens author guidelines for protocols which does not include a discussion, but ethics and dissemination after methods, no discussion or conclusion.

<https://bmjopen.bmj.com/pages/authors/#protocol>

- Limitations should be included (ex. Loss to follow-up and bias; ability to track deaths)

A: We followed BMJ Opens author guidelines for protocol which does not include a discussion, but ethics and dissemination after methods, no discussion or conclusion.

<https://bmjopen.bmj.com/pages/authors/#protocol>

Figure 1

A: Thank you for these very detailed and useful comments – both Fig1 and 2 have been updated

- In the first column it says “COVID-19 onset”. Is this true? If so, it should be clear in the text of the manuscript. If it is true, it will be hard to track things like treatment and outcomes of acute care.

- In the first column is says Medical records. This is the first instance of mentioning obtaining data from medical records.

A: the text and figure 1 has been updated to clarify how people can join the study even if they did not take part in the acute phase data collection/study.

- Column 2 mentions sampling. What type of sampling? This is not mentioned with respect to Tier 1 in the manuscript.

- The arrow points to the right. Will there be more assessments beyond that done at 12 months?
 - A: Figure 2 updated to clarify that sites can continue to follow up people at 3 to 6 months intervals depending on need and resources. This will be reviewed after interim data analysis, depending on persistence of sequelae reported.
 - This is unclear in the manuscript and this diagram tells me that assessments might continue.
 - A: Figure 2 updated to clarify that sites can continue to follow up people at 3 to 6 months intervals depending on need and resources. This will be reviewed after interim data analysis, depending on persistence of sequelae reported.
 - The term COVID-19 is written inconsistently in Figure 1 and Figure 2.
- A: Thank you for these comments – the text and figure 1 have been revised and updated to address the comments above.

Figure 2

- I found this very confusing. I am uncertain why there are 2 parallel pathways. Don't all patients have to be both confirmed as having COVID-19 and being in the ISARIC cohort? This would make 1 pathway.
 - "Confirmed COVID-19 patients" comes after "Recovered COVID-19 patients". What does this mean? Doesn't a patient have to have confirmed COVID-19 before recovering from it?
 - Where is Tier 2? Does everyone enter Tier 2? Maybe this should split off after Tier 1?
 - The open bubbles with "Immunology/Virology/Microbiomes" etc. – these have not been mentioned in the manuscript. If there is too much detail for the manuscript, a list of possible Tier 2 studies should be added as an online extra.
 - Visually, it looks like the Integrated Data Platform goes to the open bubbles. Either put it below the other bubbles or add arrows. The flow chart might be best if it was laid out in portrait format.
- A: Thank you for these very detailed and useful comments – both Fig1 and 2 have been updated for clarity to address these comments.

VERSION 2 – REVIEW

REVIEWER	Lisa Sheehy Bruyère Research Institute, Canada
REVIEW RETURNED	22-Dec-2020

GENERAL COMMENTS	The revised manuscript is much easier to follow and the figures have been nicely revised as well. There is quite a bit of repetition, which could be eliminated. A few specific things:  - Please add the term "cohort" to your study design sentence. - You mention that there are links to the surveys. I could not see them. - Line 271 "Specific Inclusion Criteria" – are these for the original acute care study or for the follow-up study? Please clarify. - The information included in your reply re. data sharing should be included in the manuscript. Most researchers and ethics review boards have regulations about where/how data is stored and shared and will need to know this information. - Figure 1 – much easier to read. Please add arrows to indicate the flow in the top section. I don't believe that the term "Tier 2" is included in the text of the manuscript now, but it is included in the figure. Please ensure consistent use of the term.
---

	- Figure 2 – much easier to read. Should “> 3-6” be “between 3-6”? Be consistent with COVID-19 (in the figures it is written as Covid-19)
REVIEWER	Ali Gholamrezanezhad Keck School of Medicine, University of Southern California (USC).
REVIEW RETURNED	13-Dec-2020
GENERAL COMMENTS	I had the pleasure to review the manuscript entitled "What is the recovery rate and risk of long-term consequences following a diagnosis of COVID-19? – a harmonised, global longitudinal observational study protocol".

VERSION 2 – AUTHOR RESPONSE

Reviewer: 1

Dr. Lisa Mary Sheehy, Bruyère Research Institute

Comments to the Author:

The revised manuscript is much easier to follow and the figures have been nicely revised as well. There is quite a bit of repetition, which could be eliminated.

A: Thank you for your thorough review. We have reviewed the manuscript to reduce repetition where appropriate throughout to make it as succinct as possible. Please find responses to the specific items spotted below.

A few specific things:

- Please add the term "cohort" to your study design sentence.

A: This has been added.

- You mention that there are links to the surveys. I could not see them.

A: Thanks for pointing out, the website link has been inserted.

- Line 271 "Specific Inclusion Criteria" – are these for the original acute care study or for the follow-up study? Please clarify.

A: The inclusion criteria is for the follow up cohort study, as the forms are developed for adults and young people from 16 years old. This has now been clarified:

Specific inclusion and exclusion criteria for the follow up cohort are as follows:

- The information included in your reply re. data sharing should be included in the manuscript. Most researchers and ethics review boards have regulations about where/how data is stored and shared and will need to know this information.

A; This has been added to the Data sharing section: Data entered into the ISARIC database are stored on the servers administered by University of Oxford Medical Sciences Division Information Technology Services (MSD ITS). Data are stored in two active primary data storage locations (Data Centres) which host mirrored copies of data, one located near the MSD ITS offices at the John Radcliffe Hospital and the second at the Old Road Campus Research Building (sites A and B respectively, both in Headington, Oxford, UK).

- Figure 1 – much easier to read. Please add arrows to indicate the flow in the top section. I don't believe that the term "Tier 2" is included in the text of the manuscript now, but it is included in the figure. Please ensure consistent use of the term.

A: The reference to Tier 2 has been clarified in the manuscript on line 294, and arrows added as suggested to Fig. 1.

- Figure 2 – much easier to read. Should "> 3-6" be "between 3-6"? Be consistent with COVID-19 (in the figures it is written as Covid-19)

A: It should be > 3 to 6 months, so that it does not overlap with the first 1 to 3 months follow up. The capitalisation of COVID-19 has been standardised in the figures, thanks for highlighting.